# The Structure and Diversity of Nitrogen Functional Groups from Different Cropping Systems in Yellow River Delta

**DOI:** 10.3390/microorganisms8030424

**Published:** 2020-03-17

**Authors:** Huan He, Yongjun Miao, Lvqing Zhang, Yu Chen, Yandong Gan, Na Liu, Liangfeng Dong, Jiulan Dai, Weifeng Chen

**Affiliations:** 1Environment Research Institute, Shandong University, Qingdao 266237, China; xina0421@163.com (H.H.); miaoyj3@mail2.sysu.edu.cn (Y.M.); sdugyd@aliyun.com (Y.G.);; 2State Key Laboratory of Agricultural Microbiology and College of Life Science and Technology, Huazhong Agricultural University, Wuhan 430070, China; 3Department of Biological Sciences, University of Essex, Wivenhoe Park, Colchester CO4 3SQ, UK; 4College of Resources and Environment, Shandong Agriculture University, Tai’an 271018, China

**Keywords:** saline–alkaline, nitrogen–cycling genes, soil bacterial community, environmental variables

## Abstract

The Yellow River Delta (YRD) region is an important production base in Shandong Province. It encompasses an array of diversified crop systems, including the corn–wheat rotation system (Wheat–Corn), soybean–corn rotation system (Soybean–Corn), fruits or vegetables system (Fruit), cotton system (Cotton) and rice system (Rice). In this study, the communities of ammonia oxidizer–, denitrifier– and nitrogen (N)–fixing bacteria in those cropping systems were investigated by Illumina Miseq sequencing. We found that Rice soil exhibited significantly higher diversity indices of investigated N–cycling microbial communities than other crop soils, possibly due to its high soil water content. Wheat–Corn soils had higher abundances of nitrification gene *amoA* and denitrification genes *nirK* and *nirS*, and exhibited higher soil potential nitrification rate (PNR), compared with Soybean–Corn, Cotton and Fruit soils. Consistently, redundancy analysis (RDA) showed that soil water content (SWC), electrical conductivity (EC), and total nitrogen (TN) were the most important influencing factors of the diversity and structure of the investigated N–cycling microbial.

## 1. Introduction

The Yellow River Delta (YRD) is one of the fastest growing deltas in the world, with great potential for land exploitation and utilization [1]. Soil salinity and alkalinity in the YRD has limited the land exploitation and agricultural development [2]. In order to alleviate the negative effects of salt–alkaline and to improve crop production, considerable nitrogen (N) fertilizers (about 1.6 × 10^5^ ton/year) have been used in YRD [3]. However, only 27–40% of applied N is taken up by crops [4]. The retained N is lost in soil through nitrate leaching, runoff and N oxides emission [5], resulting in nitrate contamination and green house pollution [6,7,8]. Therefore, improving N use efficiency has become a significant concern for agricultural sustainable development in the YRD.

Soil N use efficiency is closely related to N–cycling associated microbial communities that perform N–fixation, nitrification and denitrification [9]. Nitrogen fixers convert nitrogen gas into biologically assimilable compounds that can be taken up by crops. In intercropping systems, the nitrogen activeness of soybean nodule bacterium can reduce N fertilizer input and consequently increase N use efficiency [10]. Ammonia–oxidizer and nitrifier oxidize ammonia into nitrite and then into nitrate, whilst the denitrifier reduces nitrate into nitrite, nitrous oxide, nitric oxide, and finally nitrogen gas [11,12]. Nitrification and denitrification lead to nitrate leaching and N oxides emission [13]. Thus, suppressing ammonia-oxidizer growth was shown to increase N use efficiency of potato tubers [14]. Together, the diversity, richness and composition of N–cycling associated microbial communities affect nitrogen availability to crop and N loss from the agricultural ecosystem [15,16].

Furthermore, the diversity, richness and composition of N–cycling associated microbial communities are regulated by the cropping systems. For example, rice rotation leads to lower levels of diversity in the denitrifying community than continuous rice cropping in Japan, although denitrifying activities do not significantly vary between the two cropping systems [17]. The Rice–Wheat–Corn rotation soil supports a larger population of ammonia oxidizing archaea than the continuous corn soil [18]. However, those studies investigated only a single process of the N cycles that was affected by the cropping system. Less is known how interrelated steps of N cycle process are affected by different crop systems.

In this study, we aimed to investigate how different cropping systems affect the communities of ammonia oxidizer–, denitrifier– and nitrogen (N)–fixing bacteria. Crop systems of YRD have been progressively converted from continuous cotton to soybean system and then to wheat–corn rotation in the past two decades, and now, the cropping systems in YRD are diversified, including winter wheat-summer maize rotation system (Wheat–Corn), continuous cotton system (Cotton), and continuous rice (Rice), soybean–maize rotation system (Soybean–Corn), fruits and vegetables system (Fruit). In this study, we collected soil samples from the five cropping systems, Wheat–Corn, Soybean–Corn, Cotton, Rice, and Fruit to study the communities of total bacteria and N–cycling microbes based on the Illumina MiSeq sequencing of functional genes. We used representative primer sets to amplify 16S rDNA for total soil bacteria, *nifH* for targeting biological N–fixation bacteria, *nirS* and *nirK* for denitrifying bacteria, and *amoA* for ammonia–oxidization archaea (AOA) and ammonia–oxidization bacteria (AOB), respectively. The objectives of this study were to: (1) investigate the effects of different cropping systems in YRD on soil bacteria and targeted N–cycling functional groups; (2) investigate the contribution of soil properties to the differences in targeted N–cycling functional groups in the five cropping systems.

## 2. Materials and Methods

### 2.1. Study Area

This study was conducted in the YRD (36°55′–38°10′ N, 118°07′–119°10′ E), which is located on the south bank of the Bohai Sea, China, with a warm temperate continental monsoon climate and an average annual temperature of 12.8 °C. This delta consists of new land with unique characteristics caused by specific conditions: (1) the average yearly precipitation is 550–600 mm, in which the evaporation is 3.5 times higher (1885 mm) than that of the rainfall, resulting in difficulties with soil desalination; (2) groundwater is directly affected by the infiltration of the sea tide and the underground lateral replenishment of seawater; and (3) the land consists of a low and flat topography leading to poor drainage. All these factors have caused salt–alkalization in this region [19]. Corn, cotton, wheat, rice, and soybean constitute the main crop types in this region.

### 2.2. Soil Sampling and Preparation

Five cropping systems were selected in our study: corn-wheat rotation (*Zea mays* L.–*Triticum aestivum* L.), soybean-corn rotation (*Glycine max* L.–*Zea mays* L.), cotton (*Gossypium hirsutum* L.), rice (*Oryza sativa* L.) and fruits or vegetables (including strawberry: *Fragaria × ananassa* Duch., and Pepper: *Capsicum annuum L.*). Five samples of topsoil (0–20 cm) were collected in 30 × 30 m square diagonal lines and completely mixed as a single sample at each site. In total, corn–wheat rotation in 25 samples (Corn–Wheat, *n* = 5 sites), soybean–corn rotation in 25 samples (Soybean–Corn, *n* = 5 sites), cotton 20 samples (Cotton, *n* = 4 sites), fruit or vegetables in 20 samples (Fruit, *n* = 4 sites), and rice in 20 samples (Rice, *n* = 4 sites), were collected across the delta in June 2017. The corn–wheat rotation system, soybean–corn rotation system, cotton system, fruits or vegetables system, and rice system have been planted for approximately 7, over 20, 20, 7 and over 20 years, respectively (Figure 1). Information on fertilizer utilization is shown in Appendix A. Samples were placed into an ice box and brought to the laboratory immediately after collection. One portion of the samples was air–dried at 25° C, grinded in a mortar and sieved through successively finer meshes to obtain a 2 mm fraction for the determination of soil pH, a 0.25 mm fraction for analysis of electrical conductivity (EC), total nitrogen (TN) and ammonium contents (NH4^+^–N), nitrate concentrations (NO_3_^−^–N), available phosphorus (AP) and available potassium (AK), and a 0.149-mm fraction for measurement of effective cation exchange capacity (CEC) and soil organic matter (SOM). Other portions were frozen at −80 °C until further DNA extraction.

### 2.3. Analysis of Soil Properties

Soil pH and EC were measured in a 1:2.5 water suspension mixture using a glass electrode pH meter (PHS–2F, INESA, Shanghai, China). The soil water content (SWC) was calculated based on the losses of 20.00 g of fresh soil dried to a constant weight in an oven at 105° C [27]. SOM was determined using the Walkley–Black method [28]. The contents of NH_4_^+^–N and NO_3_^−^–N in the soil were extracted with 2 M KCl and determined on a continuous segmented flow analyser (AutoAnalyzer Ⅲ, SEAL Analytical, Fareham, UK). Soil TN was detected via the semi-micro Kjeldahl digestion method (Automatic Kjeldahl Apparatus K9860, Hanon, Jinan, China) [29]. Soil AP was estimated through extraction with sodium bicarbonate and measured on a spectrophotometer (UV –5500PC, Shjingmi, Shanghai, China) [30]. AK was extracted with 1 M ammonium acetate and measured on an Inductively Coupled Plasma–Optical Emission Spectrometer (iCAP–7000, Thermo Fisher, Waltham, MA, USA). The value of CEC was determined using a solution of barium chloride (ISO 11260–1994). To control the analytical quality, reference soil (GBW07413a) from the National Research Centre for the Certified Reference Materials of China was used in the detection processes of all indices [31]. The detected values of the standard samples fell within their accepted ranges, with a deviation of less than 5%.

### 2.4. Potential Nitrification Rate (PNR)

Potential nitrification rate was measured using 5 g of fresh soil incubated in 20 mL of pH 7.4 phosphate–buffered saline (PBS) (8.0 g/L NaCl, 0.2 g/L KCl, 1.44 g/L Na_2_HPO_4_, 0.24 g/L KH_2_PO_4_) adding 0.132 g/L (NH_4_)_2_SO_4_ and 0.05 g/L NaClO_3_ [32]. Then, NO_2_-N was extracted after shaking at 180 rpm for 0 h, 6 h, 12 h and 24 h. After filtration of the sterile filter membrane (0.45 μm), the concentration of NO_2_–N was measured using an automatic continuous flow analyser (Westco Scientific Instruments, Inc., Rome, Italy) at 550 nm.

### 2.5. Soil DNA Extraction and Real-Time Quantitative PCR

DNA was extracted from 0.25 or 0.5 g (Rice) of fresh soil using the DNeasy PowerSoil Kit (MO BIO Laboratories, Carlsbad, CA, USA) following the manufacturer’s protocol. The quality of extracted DNA was examined on a Nano Drop^®^ Spectrophotometer ND–1000 (Thermo Fisher Scientific, Waltham, MA, USA), and it was then stored at −20 °C before amplification.

The copy numbers of the five N–cycling genes were determined by Real–time quantitative PCR using FTC–3000 (Funglyn Biotech Co., Richmond Hill, Ontario, Cananda). The abundance of AOA–*amoA*, AOB–*amoA*, *nirK*, *nirS* and *nifH* genes were quantified with Arch–amoAF/Arch–amoAR, amoA–1F/amoA–2R, FlaCu/R3Cu, cd3aF/R3cd, and pol–F/pol–R, respectively (Table 1).

Real–time quantitative PCR was measured using the SYBR Premix Ex Taq^TM^ kit (Takara Bio Inc., Dalian, China) in a 25 μL reaction system consisting of 12.5 μL 2× SRBR Premix Ex Taq^TM^, 2 μL forward and reverse primers (10 μM), 5 μL template DNA and 5.5 μL RNase–free H_2_O. The AOA-*amoA* gene was amplified under the following conditions: initial melting step at 95 °C for 30 s, 40 cycles of 95 °C for 10 s, 50 °C for 30 s, and 72 °C for 40 s. The AOB–*amoA*, *nirS* and *nifH* genes was amplified under the same conditions: initial melting step at 95 °C for 30 s, 40 cycles of 95 °C for 10 s, 55 °C for 30 s, and 72 °C for 30 s. The *nirK* gene was amplified as follows: initial melting step at 95 °C for 30 s, 40 cycles of 95 °C for 10 s, 60 °C for 30 s, and 72 °C for 30 s. Each sample had three replicates. Standard curves were created by seven–fold serial dilutions of plasmids containing the corresponding gene. The linear coefficient correlation (*R*^2^) of the standard curves were 0.9998 for AOA–*amoA*, 0.9999 for AOB–*amoA*, 0.9999 for *nifH*, 0.9992 for *nirK*, 0.9992 for *nirS*.

### 2.6. High–Throughput Sequencing and Data Analysis

The V3–V4 regions of bacteria (16S rRNA) and five N–cycling genes (*amoA*–AOA, ammonia oxidation archaea; *amoA*–AOB, ammonia oxidation bacteria; *nirS* and *nirK*, nitrite reduction bacteria; *nifH*, N–fixation bacteria) were amplified using different primer pairs targeting specific genetic regions or genes. The amplification primer pairs and thermal cycling conditions of bacteria 16S rRNA and functional N–cycling genes are listed in Table 1. Each 50 μL PCR reaction system was prepared using 10 μL 5x Buffer, 1 μL dNTP (10 mM), 1 U Phusion^®^ High–Fidelity DNA Polymerase, 1 μL (10 μM) each of forward and reverse primers, and 5–50 ng of template DNA supplemented with ddH_2_O. The PCR products were purified using the High Pure PCR Product Purification Kit (Roche Applied Science, Meylan, France). Purified PCR products were well mixed to establish sequencing libraries after further qualification, and the libraries were sequenced on an Illumina MiSeq platform (TinyGene Inc., Shanghai, China).

Raw data were filtered by Trimmomatic (V 0.32). Reads were truncated when the average quality score was less than 20 over a 50–base window. Trimmed reads shorter than 50 bp were also discarded. Then, forward and reverse reads were merged using FLASH with a 10 bp minimum overlap and a maximum mismatch density of 0.2. Mothur (V 1.39.5) was used to filter the reads that were ambiguous and homologous with the parameters: maxambig = 0, maxhomop = 8, minlength = 200 (for 16S rRNA, *amoA*–AOB, *nifH*, *nirK* and *nirS*)/150 (for *amoA*–AOA), maxlength = 580. Operational taxonomic units (OTUs) with a 97% similarity cutoff were clustered using UPARSE (V 7.1) and chimeric sequences were removed by UCHIME. The taxonomy of 16S rRNA was predicted against the SILVA 128 database using Mothur (V 1.39.5); the taxonomies of *amoA*, *nifH*, *nirK* and *nirS* were selected against the NCBI database. The differences in diversity and richness of the bacterial and chosen N–cycling functional groups amongst the five cropping systems were examined by using multiple independent sample non–parametric tests (Kruskal–Wallis H test). For β–diversity, to determine the differences of soil microbial community structure amongst different cropping systems, we performed Non–Metric Multidimensional Scaling (NMDS) using the Multiple Response Permutation Procedure (MRPP) from the library “vegan” in R (V 3.5.0).

To quantify the relationship between soil properties and bacteria and N–cycling functional genera, Spearman correlation coefficients were calculated using SPSS Statistics (V 19.0). We performed a redundancy analysis (RDA) based on the soil properties and functional N–cycling microbes at the genus level to determine the main environmental factors affecting the N–cycling processes in each cropping system using the Canoco software (V 5.0).

To investigate the relationship of dominant OTUs amongst the five cropping systems, Co–occurrence networks were performed. Firstly, we defined the “dominant” OTUs with relative abundances > 0.5% within at least one replicate in each cropping system. Then, the matrix was established based on the Pearson correlation coefficient using the relative abundance of each “dominant” OTU. The network was conducted through employing a threshold (Pearson correlation coefficient |*r*| > 0.85, *p* < 0.05). The network structure was visualized with Cytoscape (V 3.6.0). In the network structure, node values represent the relative abundances of OTUs, and the edge sizes indicate the Pearson correlation coefficients between each two OTUs. And the network parameters including the number of edges, the number of nodes, clustering coefficients, and network density were identified using the Network Analysis method.

### 2.7. Accession Numbers

Nucleotide sequence data were submitted to the National Centre for Biotechnology Information (NCBI) Sequence Read Archive (SRA) with accession number SRP152899.

## 3. Results

### 3.1. Soil Physicochemical Properties

Differences between the soil physicochemical properties of the five cropping systems were shown in Table 2. Rice soils contained a significantly higher (*p* < 0.001) concentration of SWC, reaching 38.1 ± 4.4% compared to the other four cropping systems. This could be explained by the higher water content of rice habitats compared with other crops. The highest AP (71.7 mg kg^−1^) and AK (0.37 g kg^−1^) were found in Fruit soils, and significantly differed (*p* < 0.05) from others. There were no significant differences in the contents of SOM, TN, NH_4_^+^–N, NO_3_^−^–N, EC and CEC amongst the five cropping systems.

### 3.2. Diversity of Bacteria and N–Cycling Functional Groups in the Five Cropping Systems

*α*–*diversity:* The coverage of all tested soil samples was more than 99% (Table 3), which indicated that there was a sufficient level of Illumina MiSeq sequencing to identify most diversity of soil samples. Rice soils had the highest OTUs of bacteria, followed by ammonia–oxidizer, denitrifier and N–fixing bacteria. The Chao and ACE indices showed a similar pattern to those of the OTUs, i.e., rice soils had the highest Chao and ACE indices of ammonia–oxidizer, denitrifier and N–fixing bacteria, whilst no significant differences in the Shannon index existed amongst the five cropping systems.

*β*–*diversity:* The compositions of bacteria, ammonia-oxidizers and denitrifying bacteria in rice soils differed to those of other soils, and no overlap was present based on the NMDS analysis (Figure 2a–c). This was supported by the Multi Response Permutation Procedure (MRPP) analysis (Table 4), showing that the community compositions of soil bacteria, ammonia oxidizer *amoA*–AOA and denitrifier *nirS* significantly differed amongst the five cropping systems. The community composition of N–fixing bacteria in the five cropping systems had a large area of overlap (Figure 2d), indicating that there was no difference. Additionally, MRPP showed no significant differences in the community composition of N–fixing bacteria amongst the five cropping systems.

### 3.3. Composition and Difference in the Bacterial Community

A total of 55 phyla were identified in all tested soils. The phyla *Proteobacteria*, *Acidobacteria*, *Chloroflexi* and *Gemmatimonadetes* were the most abundant phyla (> 5% of all sequences in each cropping system) in the soil of Wheat–Corn, Cotton, Fruit and Soybean–Corn, with the range of 30.5–33.7%, 11.0–23.1%, 10.2–13.5% and 5.1–10.3%, respectively (Figure 3A). Moreover, we found that the relative abundance of *Bacteroidetes* was significantly higher, whilst the *Nitrospirae* was significantly lower in Rice soils compared with other soils (Figure 4).

Figure 3B showed the relative abundance of the 50 most abundant bacterial genera in the five ecosystems. Bacterial genera from Fruit, Cotton, Soybean–Corn and Wheat–Corn clustered into a single group. Rice soils were separated from other four site samples. The composition of the soil bacterial community at the genus level in the rice samples relatively differed from the other samples, in line with the NMDS and MRPP analyses.

### 3.4. Abundance of N–Cycling Genes

The abundance of N–cycling genes, including AOA–*amoA*, AOB–*amoA*, *nirK*, *nirS* and *nifH* genes were determined. As shown in Figure 5, the log 10 of the five genes’ copy number ranged from 5.8 to 7.2 g^−1^ soil for AOA–*amoA*, 6.5 to 7.2 g^−1^ soil for AOB–*amoA*, 5.1 to 6.1 g^−1^ soil for *nirK*, 6.5 to 7.3 g^−1^ soil for *nirS*, 5.8 to 7.3 g^−1^ soil for *nifH*, respectively. Wheat–Corn soils had significantly higher abundances of ammonia oxidizer *amoA* and denitrification *nirK*, whilst Rice soils had lower abundance of AOA–*amoA* and higher denitrification genes (*nirK* and *nirS*) abundance, compared to the Cotton, Fruit and Soybean–Corn soils. The abundances of *nifH* were significantly higher in Rice samples than those of the other four cropping system soils (*p* < 0.05).

### 3.5. Potential Nitrification Rate (PNR)

The PNR of the different cropping system soils was shown in Figure 6. Soil PNR showed no significant differences in Wheat–Corn, Soybean–Corn, Cotton and Fruit. However, the soil PNR was lower in Rice soils (*p* < 0.05). Generally, PNR represents the ability of ammonia oxidizer (AOA and AOB) in the soil to transform NH4^+^–N to NO_2_^−^–N. Therefore, lower PNR values indicate a lower activity of soil ammonia oxidizer in Rice soils, compared to the other four cropping systems.

### 3.6. Composition of N–Cycling Functional Communities

Of the genera in the well-identified AOA groups, *Nitrososphaere* from phylum Thaumatchaeota was the most abundant genus in the five agroecosystems, with a content of 25.3–29.9%, followed by *Nitrosopumilus* (3.6–16.1%) and *Candidatus_Nitrosocosmicus* (0–3.4%) (Table 5). For AOB members, *Nitrosospira*, *Nitrosomonas*, *Nitrosovibrio* and *Roseateles* were the four most abundant genera in the five agroecosystem soils, accounting for 43.2–81.7%, 1.8–40.0%, 2.1–8.1% and 0–1.9%, respectively (Table 5).

For nitrogen–fixing bacteria (*nifH*), *Azospirillum*, *Paraburkholderia*, *Geobacter*, *Skermanella*, *Desulfuromonas*, *Desulfovibrio*, *Azotobacter* were the dominant genera (> 5% of all sequences in at least one agroecosystem), accounting for 1.7–20.1%, 1.2–9.2%, 3.3–6.9%, 0.2–7.1%, 1.6–7.7%, 2.2–6.2%, 1.6–7.7%, 0.6–9.0%, 0.5–5.4% of all sequences, respectively (Table 5).

In *nirK*–type denitrifying bacteria, *Mesorhizobium*, *Sinorhizobium*, *Rhizobium* and *Bradyrhizobium* were the most abundant genera (> 5% of all sequences in at least one agroecosystem) in the five agroecosystem soils, accounting for 12.5–20.4%, 6.2–22.9%, 6.5–29.1% and 10.7–15.9%, respectively (Table 5). Additionally, the genera *Azoarcus*, *Rhodanobacter*, *Cupriavidus*, *Rubrivivax*, *Azospirillum*, *Pseudomonas* and *Sulfuritalea* were the most abundant members (> 5% of all sequences in at least one agroecosystem) of *nirS*–type denitrifying bacteria, with abundances of 3.3–13.1%, 3.3–16.9%, 2.2–6.9%, 1.6–5.6%, 0.9–7.0%, 0.8–9.7%, 1.9–6.8%, respectively (Table 5).

### 3.7. Co–Ocurrence Network Analysis of amoA–AOA and nirS–Denitrifier

The first stage of nitrification involves the oxidation of ammonia to nitrite, which is catalysed by ammonia monooxygenase (AMO) [23]. The genes *nirS* and *nirK*, as marker genes of denitrifier [33], catalyse the reduction of nitrite to nitric oxide. Nitrification and denitrification are important activities in during the loss of ammonium-based fertilisers [34,35]. Based on MRPP analysis, the compositions of ammonia oxidizer *amoA*–AOA and *nirS* denitrifier were significantly different amongst the five cropping system soils (Table 4). A network analysis was therefore used to explore the differences of *amoA*–AOA and *nirS*–type denitrifier communities in the different cropping systems (Figure 7 and Figure 8), to a better understand N–cycling functional community structures.

The network parameters of the two communities were shown in Appendix A. The node and edge of *amoA*–AOA in the Wheat–Corn and Cotton soils were higher than those of other cropping systems. The edges of *nirS*–denitrifier in Fruit soils were higher than those of other soils. However, the network density of *amoA*–AOA in Wheat–Corn soil was higher than that of other soils. For the *nirS*–type denitrifier networks, the network density in Rice was higher than that of other soils.

### 3.8. Key Drivers of Bacterial and N–Cycling Functional Communities

In this study, Rice soils were well separated through RDA analysis (Figure 9), indicating that the composition of the chosen N–cycling functional groups in Rice soil differed to those of other soils. Moreover, RDA analysis showed that the environmental factors, SWC, EC, TN, AP and CEC were important and influential factors of bacterial and the chosen N–cycling functional groups (Table 6). Amongst them, SWC could explain 21.6–54.2% of the variation in the functional communities of different cropping systems, and was the most important influencing factor (Table 6).

EC and TN were also related to the ammonia oxidizer AOA and AOB; EC and AP were related to denitrifier *nirS* and *nirK*; and CEC was related to *nifH*. EC explained 15.3% and 13.7% of the variation of ammonia oxidizer and denitrifier compositions, respectively. TN, AP and CEC were minor drivers of bacterial and N-cycling functional communities, which were related to ammonia oxidizer, denitrifier and *nifH*–type fixing bacteria, respectively.

## 4. Discussion

### 4.1. Abundance and Diversity of Chosen N–Cycling Functional Genes in Different Cropping Systems

In this study, we investigated the communities of soil bacteria, ammonia oxidizer, denitrifier and nitrogen–fixing bacteria by high throughput sequencing. High throughput sequencing reveals the presence of organisms containing these genes but their viability and metabolic activation have not been assessed. Nevertheless, the method has been more widely used to analyze microbial communities than cultivation-dependent method, as less than 1% of the bacterial taxa can be cultivated [36].

Here, we demonstrate that Wheat–Corn rotation soil had the most abundant nitrification gene *amoA* and relatively high PNR activity. Additionally, the soil had the highest microbial network density of *amoA*–AOA. Microbial network density reflects microbial interconnectivity, which likely affect microbial functions [37]. Collectively, the data suggest that Wheat–Corn rotation soil had higher ammonia oxidization activity than other cropping systems. This phenomenon could be caused by the application of N fertilizer, since the abundance of ammonia oxidizer positively correlated with ammonia availability in agricultural soil [38,39]. Consistently, Wheat–Corn rotation soil is supplemented with an average of compound fertilizer 600 kg h^−1^year^−1^ and diammonium hydrogen phosphate 550 kg h^−1^year^−1^, which is higher than other soils (Appendix A). Moreover, Wheat–Corn rotation soil has high abundances of denitrification genes (*nirK* and *nirS*), suggesting that it contains a higher number of denitrification microbes. We reasoned that the NO_3_^−^ produced by ammonia oxidizer and nitrite oxidizing bacteria, could promote the growth of denitrifier. Noticeably, a reduction in NO_3_^−^ by denitrifier leads to N_2_O, which is a powerful greenhouse gas. Thus, the excessive application of N fertilizer increases the abundance of ammonia oxidizer and denitrifer, consequently enhancing NO_3_^−^ leaching and N_2_O emission. Therefore, suppression of ammonia oxidizer would be an effective approach to increase nitrogen fertilizer use efficiency.

We further reveal that AOB was more abundant than AOA in Wheat–Corn soils. AOB is favoured by inorganic fertilization and produces high N_2_O emission, whilst AOA is favoured by organic or slow–release fertilizer, yielding lower N_2_O emission [40]. Together, we deduce that Wheat–Corn soils have higher N_2_O emissions and consequently lower fertilizer use efficiency. The application of nitrification inhibitor and slow-release fertilizer should be taken into consideration in the Wheat–Corn cropping system to suppress ammonia oxidization and decrease AOB abundance, thereby decreasing N_2_O emission and improving N use efficiency.

Rice soil contained the most abundant nitrogenase gene *nifH* and denitrification genes (*nirK* and *nirS*), due to their high water content (discussed in Section 4.2). Rice soil also exhibited higher diversity of *amoA*–AOA and *nirS* communities than other cropping systems according to ACE and Chao analysis. However, the results contrasted those of microbial network density analysis, revealing that the nodes and edges of the networks of Rice soils were less than those of Wheat–Corn, Cotton and Fruit soils. Since the number of edges reflect the degree of microbial diversity [41,42], the microbial network density analysis suggested simplified *amoA*–AOA and *nirS* communities in Rice and Soybean–Corn soils. The reason for these differences may result from the network density analysis only using “dominant” OTUs with a relative abundance > 0.5%. These data suggest that Rice soil has a higher diversity but fewer dominant species of *amoA*–AOA and *nirS* communities.

It should be realized that PCR primer design is based on current sequence databases and microorganism habitats, and that on primer sets can achieve a comprehensive coverage of natural sequences or unbiased amplifications [43]. In this study, the primer sets Arch amoA–F/Arch amoA–R for targeting AOA and amoA–1F/ amoA–2R for targeting AOB have limitations in terms of the comprehensive coverage of *Nitrosotalea* subclade 1.1 and *Nitrosospira* 8A clade, respectively. However, this may reduce the amplification bias of some minor populations of ammonia oxidizer members [44]. For *nir* genes, the conventional primers F1aCu/R3Cu and cd3aF/R3cd closely match the sequences of Cu-containing nitrite reductase (*nirK*) and cytochrome cd1 nitrite reductase (*nirS*) respectively [26,45]. The primers preferentially target subclasses of Proteobacteria [46] but fail to amplify previously unconsidered *nir* genes in newly discovered microorganisms [34]. In addition, the *nifH* primer set PolF/PolR can amplify three of the four clusters of *nifH* paralogs. The cluster IV/V of *nifH* that cannot be identified by PolF/PolR skews the diversity analysis and their sequences can be filtered before diversity analysis [47]. Moreover, the primer set PolF/PolR has lower template–specific bias in qPCR application than other primer sets, thereby more accurately evaluates *nifH* copy numbers [48]. The further development of clade–specific and habitat–specific primers is important to reveal the diversity and abundance of environmental microorganisms. Developing primers with a higher coverage and specificity of N–cycling genes in the saline–alkali soil of YRD remains a goal of further studies.

### 4.2. Factor Affecting the Diversity and Composition of Bacterial and N–Cycling Functional Communities

According to RDA analysis (Table 6), SWC was the most important influencing factor for the variation of bacterial and targeted N–cycling functional communities. The habitat with high water content possesses more functionally diverse microbial communities, maintaining the activity of soil enzymes [49,50]. Additionally, high SWC can alleviate the negative effects of soil salinization on crop growth, as the irrigation water can push salts below the root zone [51].

SWC had a significant effect on the total bacterial community (*p* < 0.05) (Table 6). Specifically, the relative abundance of bacterial phylum Bacteroidetes was positively related to SWC (Spearman’s correlation coefficients = 0.469, *p* < 0.01), and the phylum was significantly higher in Rice soil than other crop type soils (Figure 4). It is well recognised that phylum Bacteroidetes prominently inhabits water-based environments [52]. For diazotrophs, the abundance of *nifH* was higher in Rice soils than other cropping soils (*p* < 0.05) (Figure 5B), and the SWC was positively related to N fixing bacterial community. This phenomenon could be explained by the higher SWC that can enhance the diffusive transport of organic substrates, which may indirectly increase the access of organic C substrates to N fixing bacteria [11]. Furthermore, SWC has a significant effect on denitrifier [53]. The abundances of denitrification *nirK* and *nirS* were higher in Rice habitat with high SWC. That is because the rice habitat with high SWC can provide an anaerobic environment for denitrifying microbe where the abundance and diversity of denitrifying genes remains relatively high [54].

Additionally, soil TN was an important factor driving the composition of ammonia oxidizer (Table 6). The abundances of AOA and AOB were higher in Wheat–Corn soils with high fertilizer level. The results agreed with that abundances of AOA and AOB were positively related to ammonia availability in agricultural soil [38,39]. In addition, soil TN had no significant effect on the structure of N fixing bacterial communities in our study (*p* > 0.05) but could reduce *nifH* abundance. These data are consistent with that the activity of *nifH* community can be suppressed when nitrogen is abundant [55].

Here, we demonstrate that the studied N–cycling microbial communities were affected by the cropping strategy and soil properties, such as SWC and TN. These factors also influence N use efficiency [13]. The results support that optimizing soil water management and nitrogen fertilizer is a strategy to improve the N use efficiency in agroecosystem of YRD. Future studies should focus on the relationship between N–cycling microbial communities and N cycle activities, such as ammonium oxidation, denitrification and N fixation in the crop systems to reveal the contribution of N–cycling microbes to the N cycle.

## 5. Conclusions

This study has clarified the effects of different cropping systems in the YRD on the abundance and diversity of selected N functional genes, including the nitrification gene *amoA*, the denitrification genes *nirK* and *nirS,* and the nitrogenase gene *nifH*. We explored the contribution of soil properties to the abundance and diversity of these genes. We found that (1) Rice soil had significantly higher diversity indices in terms of soil bacterial and targeted N–cycling functional communities than other crop soils; (2) Rice soils with higher SWC had lower *amoA* abundance and soil PNR, indicating that Rice soils possess lower abundance and activity of ammonia oxidizer; (3) Wheat–Corn soil with higher level of fertilization had higher abundance of nitrification gene *amoA* and denitrification genes *nirK* and *nirS*, compared with Soybean–Corn, Cotton and Fruit soils; (4) SWC, EC, and TN were the most important influencing factors of soil bacterial and targeted N–cycling microbial diversity and structure. Additionally, considering that compound fertilizer 600 kg h^−1^year^−1^ and diammonium hydrogen phosphate 550 kg h^−1^year^−1^ were used in the Wheat–Corn soils, urgent actions are required to improve N fertilizer use efficiency and to reduce N loss, for example, optimizing soil water management and the content of nitrogen fertilizers.

## Figures and Tables

**Figure 1 microorganisms-08-00424-f001:**
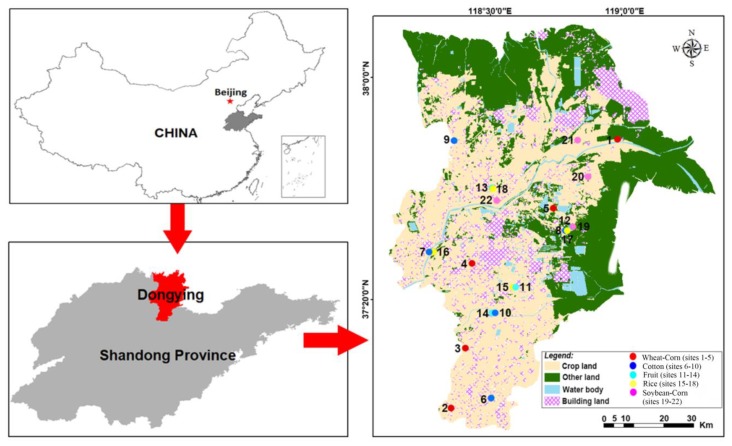
Location of the 22 sites collected from the Yellow River Delta, China according to crop type, i.e., wheat-corn rotation (Corn–Wheat, sites 1–5), cotton (Cotton, sites 6–10), fruits or vegetables (Fruit, sites 11–14), rice (Rice, sites 15–18), soybean-corn rotation (Soybean–Corn, sites 19–22).

**Figure 2 microorganisms-08-00424-f002:**
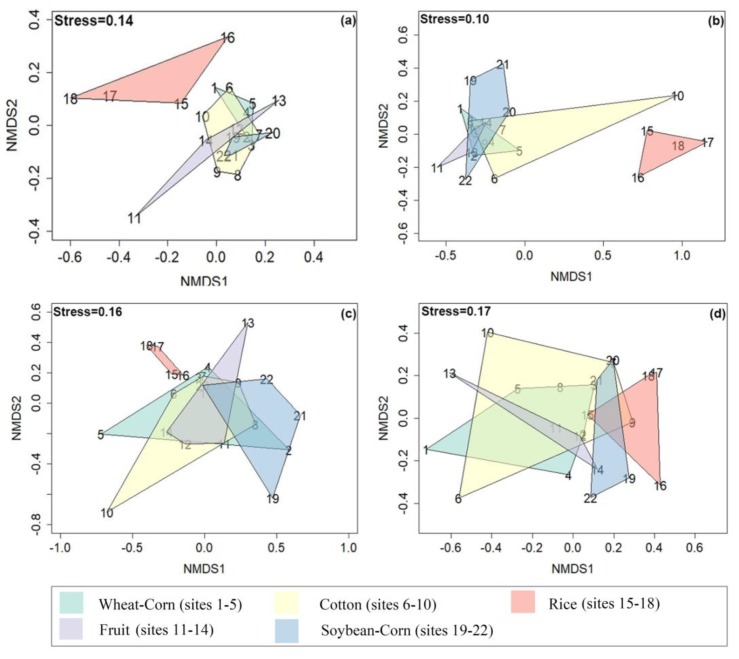
Non–Metric Multidimensional Scaling (NMDS) analysis of amplicon profiles of 16S rRNA of Bacteria (**a**), of *amoA* of ammonia–oxidizers (**b**), of *nirK* and *nirS* of denitrifying bacteria (**c**), of *nifH* of N–fixing bacteria (**d**) from different cropping systems, i.e., wheat–corn rotation (Wheat–Corn, sites 1-5), cotton (Cotton, sites 6-10), fruits or vegetables (Fruit, sites 11-14), rice (Rice, sites 15-18), soybean–corn rotation (Soybean–Corn, sites 19-22). Ordinations were based on Bray–Curtis dissimilarities.

**Figure 3 microorganisms-08-00424-f003:**
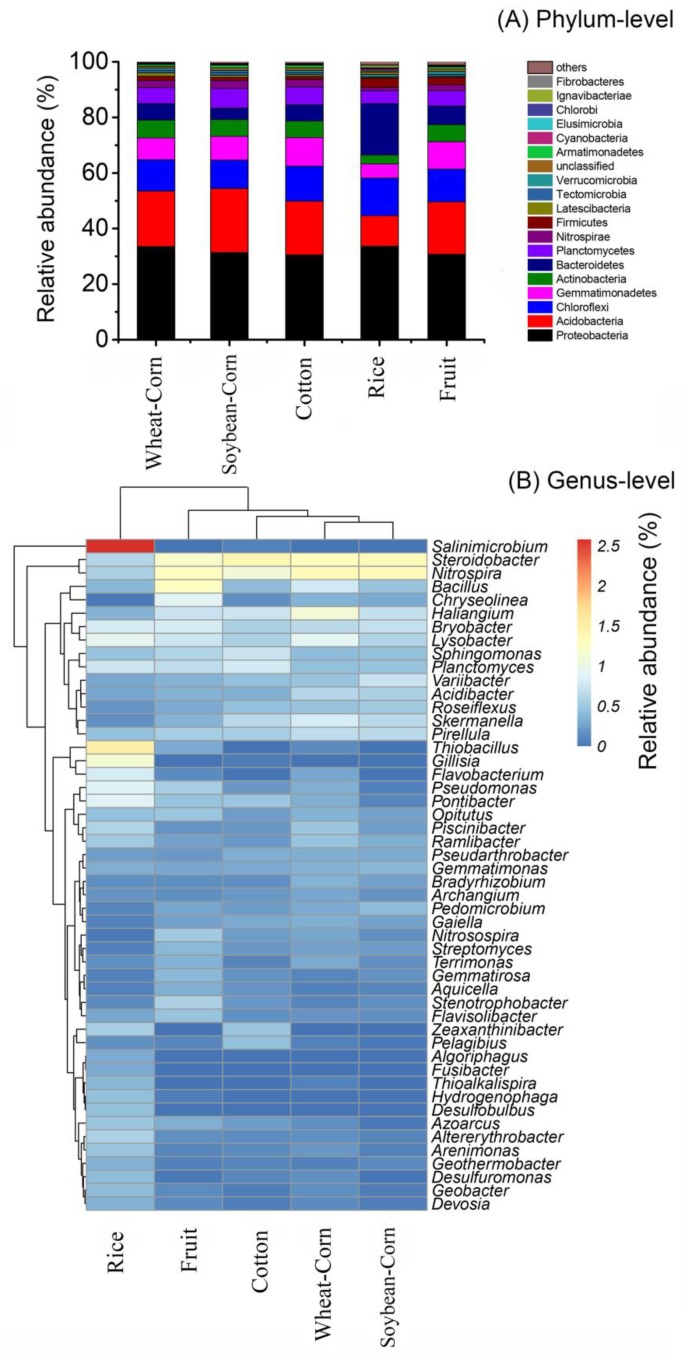
Phylum (**A**) and Genus (**B**) compositions of soil bacteria in the five cropping systems. Heatmap (B) showing the relative abundance of the top 50 abundant bacterial genera. Dendrogram on top of the heatmap shows relationship between agroecosystem types based on the relative abundance of bacterial genera using Euclidian distance.

**Figure 4 microorganisms-08-00424-f004:**
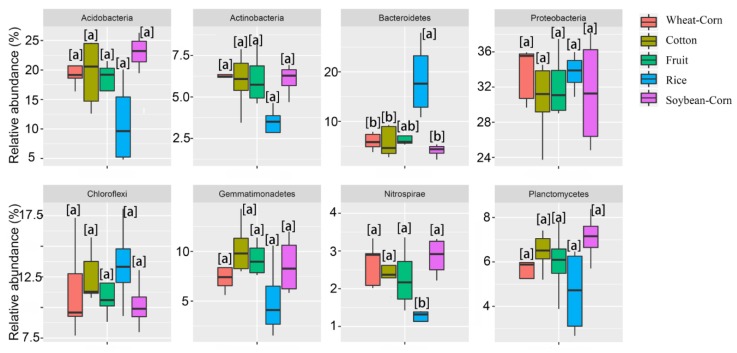
Boxplots representing the relative abundance of the eight domain bacterial phyla discovered in the five cropping systems: wheat corn rotation (Corn–Wheat), cotton (Cotton), rice (Rice), fruits or vegetables (Fruit), soybean corn rotation (Soybean–Corn). Different letters indicate significant differences based on Kruskal–Wallis test (*p* < 0.05).

**Figure 5 microorganisms-08-00424-f005:**
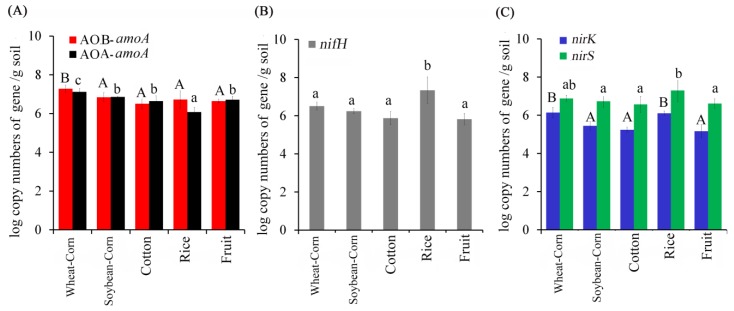
Number of AOA and AOB *amoA* gene copies (**A**), *nifH* gene copies (**B**) and *nirK* and *nirS* (**C**) in the five cropping systems: wheat corn rotation (Corn–Wheat), cotton (Cotton), rice (Rice), fruits or vegetables (Fruit), soybean corn rotation (Soybean–Corn). One–way ANOVA was used to determine the difference in the abundance of nitrogen functional genes. Statistically significant differences are noted by different letters. In Figure 5A, different capital letters indicate significant differences of AOB *amoA* abundance, while different lowercase letters indicate significant differences of AOA *amoA* abundance. In Figure 5C, different capital letters indicate significant differences of *nirK* abundance, while different lowercase letters indicate significant differences of *nirS* abundance.

**Figure 6 microorganisms-08-00424-f006:**
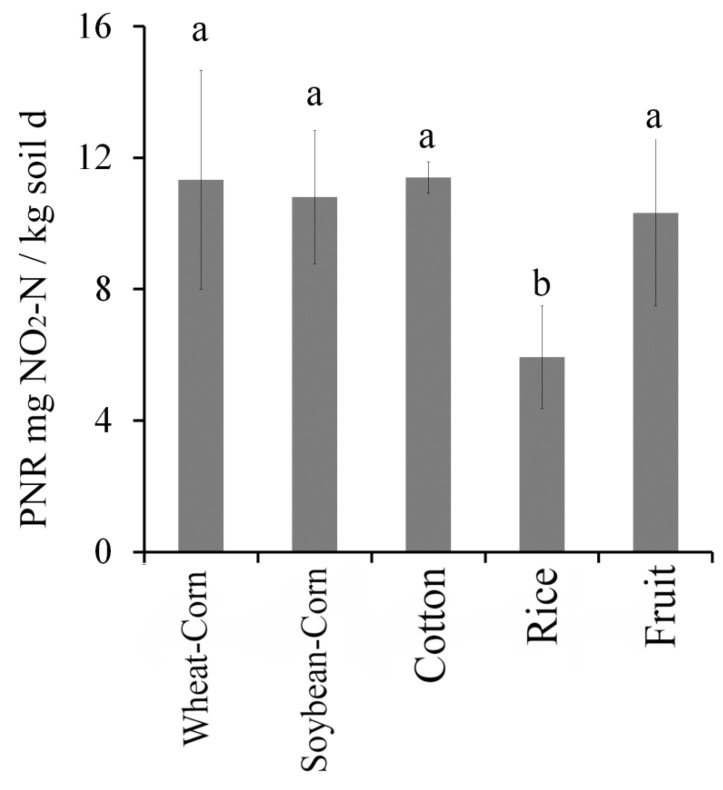
Soil PNR in the five cropping systems: wheat corn rotation (Corn–Wheat), cotton (Cotton), rice (Rice), fruits or vegetables (Fruit), soybean corn rotation (Soybean–Corn). One-way ANOVA was used to determine differences in soil PNR across the different cropping systems. Statistically significant differences are denoted by different letters.

**Figure 7 microorganisms-08-00424-f007:**
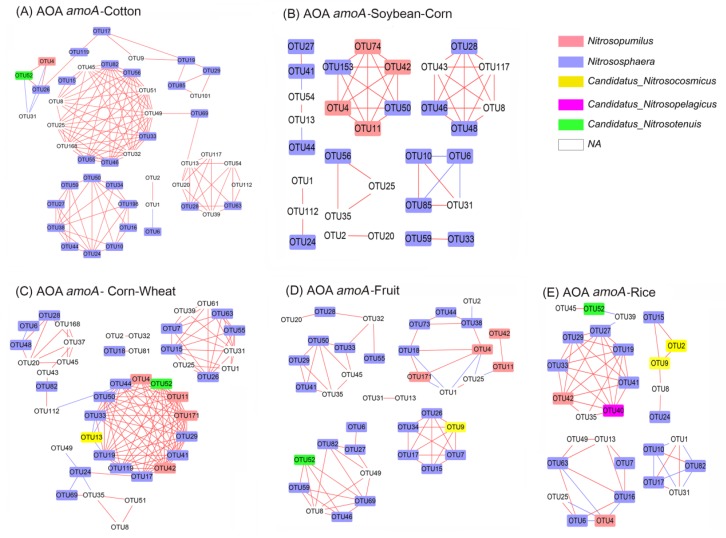
Network analysis of *amoA*–AOA OTUs in the five ecosystems: (**A**) Cotton; (**B**) Soybean–Corn; (**C**) Wheat–Corn rotation; (**D**) Fruit; (**E**) Rice. The network shows only the “dominant” OTUs with a relative abundance >0.5% in at least one replicate. Red solid lines: significantly positive relationships (Pearson correlation *r* > 0.85); blue solid line: significantly negative relationships (Pearson correlation *r* < −0.85).

**Figure 8 microorganisms-08-00424-f008:**
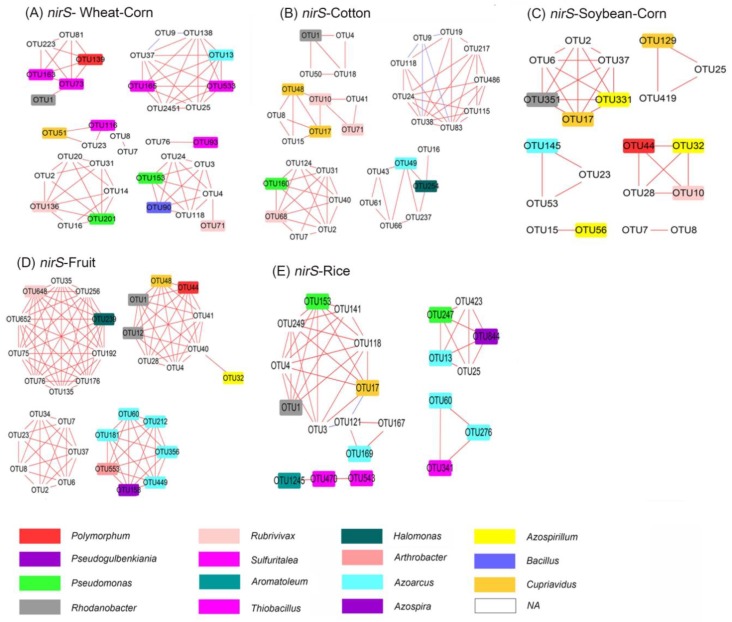
Network analysis of *nirS*-denitrifier OTUs in the five ecosystems: (**A**) Wheat–Corn; (**B**) Cotton; (**C**) Soybean–Corn; (**D**) Fruit; (**E**) Rice. The network only shows the “dominant” OTUs with a relative abundance > 0.5% in at least one replicate. Red solid lines: significantly positive relationships (Pearson correlation *r* > 0.85); blue solid line: significantly negative relationships (Pearson correlation *r* < −0.85).

**Figure 9 microorganisms-08-00424-f009:**
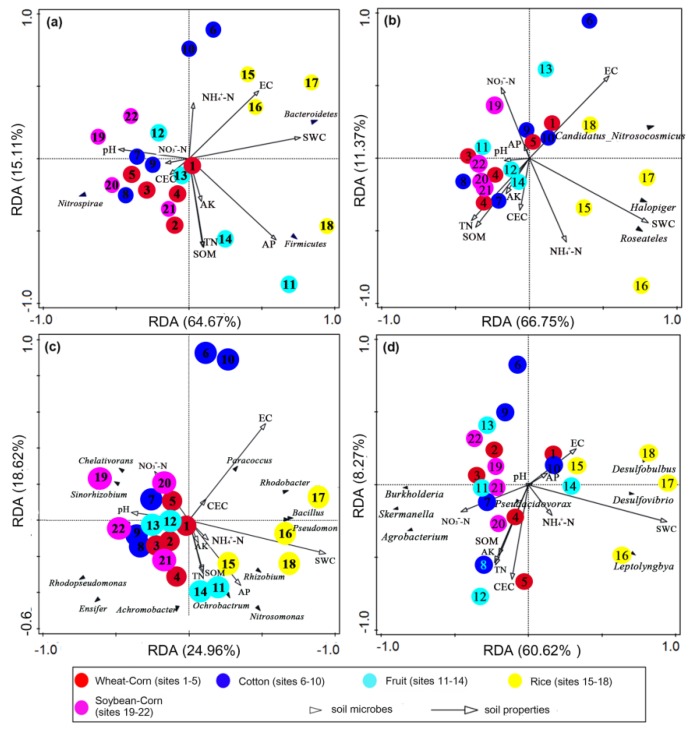
Redundancy analyses of the soil properties and (**a**) bacteria at the phylum level, (**b**) ammonia–oxidizers (*amoA*), (**c**) denitrifying bacteria (*nirS* and *nirK*), and (**d**) N–fixing bacteria (*nifH*) at the genus level in the five agroecosystem soils: corn wheat rotation (Wheat–Corn), cotton (Cotton), rice (Rice), fruits or vegetables (Fruit), soybean-based rotation (Soybean–Corn). SWC, SOM, CEC, NO_3_^−^–N, NH_4_^+^–N, TN, AK, AP, EC and pH represent the water content, organic matter, cation exchange capacity, nitrate, ammonium, total nitrogen, available potassium, available phosphorous, electrical conductivity and pH of the soil samples, respectively.

**Table 1 microorganisms-08-00424-t001:** Primer pairs used for gene detection in this study.

Genes	Premiers	Premier Sequences (5′–3′)	Reference(s)
16S rRNA	515F	GTGCCAGCMGCCGCGGTAA	Ding et al. (2015) [20]
	926R	CCGTCAATTCMTTTGAGTTT	
*nifH*	PolF	TGCGAYCCSAARGCBGACTC	Poly et al. (2001) [21]
	PolR	ATSGCCATCATYTCRCCGGA	
*amoA*–AOA	Arch amoA F	STAATGGTCTGGCTTAGACG	Francis et al. (2005) [22]
	Arch amoA R	GCGGCCATCCATCTGTATGT	
*amoA*–AOB	amoA–1F	GGGGTTTCTACTGGTGGT	Rotthauwe et al. (1997) [23]
	amoA–2R	CCCCTCKGSAAAGCCTTCTTC	
*nirS*	cd3aF	GTSAACGTSAAGGARACSGG	Michotey et al. (2000) [24]
	R3cd	AGTTCTGSGTRGGCTTSAG	Hallin and Lindgren (1999) [25]
*nirK*	F1aCu	ATCATGGTSCTGCCGCG	Kandeler et al. (2009) [26]
	R3Cu	GCCTCGATCAGRTTGTGGTT	

**Table 2 microorganisms-08-00424-t002:** The properties of soil in the corn and wheat rotation (Corn–Wheat), fruits (Fruit), rice, soybean and corn rotation (Soybean–Corn), and cotton agroecosystems used in this study. SWC, soil water content; AP, available phosphorous; AK, available potassium; pH; SOM, organic matter; TN, total nitrogen; NH_4_^+^–N, ammonium; NO_3_^−^–N, nitrate; CEC, cation exchange capacity; and EC, electrical conductivity.

Agroecosystem	SWC (%)	AP (mg kg^−1^)	AK (g kg^−1^)	pH	SOM (g kg^−1^)	TN (g kg^−1^)	NH_4_^+^–N (mg kg^−1^)	NO_3_^−^–N (mg kg^−1^)	CEC [cmol (+) kg ^−1^]	EC (mS cm^−1^)
Wheat–Corn	21.3 ± 3.7 ^a^	14.5 ± 11.1 ^a^	0.18 ± 0.05 ^a^	8.0 ± 0.0 ^a^	14.9 ± 3.3 ^a^	0.83 ± 0.17 ^a^	8.0 ± 1.2 ^a^	36.7 ± 29.4 ^a^	12.6 ± 2.3 ^a^	0.34 ± 0.16 ^a^
Soybean–Corn	16.3 ± 2.7 ^a^	10.4 ± 7.8 ^a^	0.19 ± 0.04 ^a^	8.1 ± 0.2 ^a^	14.1 ± 4.1 ^a^	0.78 ± 0.23 ^a^	9.1 ± 0.9 ^a^	32.9 ± 18.4 ^a^	10.5 ± 3.0 ^a^	0.30 ± 0.09 ^a^
Cotton	18.3 ± 1.7 ^a^	14.0 ± 9.3 ^a^	0.22 ± 0.06 ^a^	8.0 ± 0.1 ^a^	13.2 ± 3.4 ^a^	0.73 ± 0.21 ^a^	7.7 ± 0.7 ^a^	48.1 ± 30.3 ^a^	11.4 ± 1.7 ^a^	0.65 ± 0.48 ^a^
Rice	38.1 ± 4.4 ^b^	31.8 ± 20.0 ^a^	0.12 ± 0.07 ^a^	7.9 ± 0.1 ^a^	13.1 ± 2.6 ^a^	0.72 ± 0.15 ^a^	10.9 ± 4.6 ^a^	8.7 ± 4.8 ^a^	10.7 ± 3.5 ^a^	0.63 ± 0.23 ^a^
Fruit	21.0 ± 3.7 ^a^	71.7 ± 42.9 ^b^	0.37 ± 0.11 ^b^	7.8 ± 0.3 ^a^	16.2 ± 5.5 ^a^	1.00 ± 0.36 ^a^	7.8 ± 0.9 ^a^	75.1 ± 41.1 ^a^	11.6 ± 1.6 ^a^	0.60 ± 0.22 ^a^

Different letters indicate significant differences based on Kruskal–Wallis test (*p* < 0.05).

**Table 3 microorganisms-08-00424-t003:** Microbial richness indices of the samples from different agroecosystems.

Microbial Community	Gene Type	Samples	Number of OTUs	Coverage (%)	Chao Index	Ace Index	Shannon Index
Ammonia-oxidizers	*amoA*–AOA	Wheat–Corn	72.40 ± 4.80 ^a^	99.94 ± 0.02^a^	77.21 ± 4.70 ^a^	77.78 ± 5.90 ^a^	2.55 ± 0.28 ^a^
		Soybean–Corn	72.00 ± 3.00 _a_	99.94 ± 0.01^a^	79.45 ± 5.31 ^a^	80.64 ± 3.51 ^ab^	2.20 ± 0.15 ^a^
		Cotton	67.00 ± 4.73 _a_	99.94 ± 0.02^a^	75.07 ± 3.72 ^a^	77.66 ± 6.15 ^a^	2.24 ± 0.29 ^a^
		Rice	87.50 ± 2.87 _b_	99.95 ± 0.01^a^	94.33 ± 2.90 ^b^	93.33 ± 3.26 ^c^	2.55 ± 0.14 ^a^
		Fruit	75.50 ± 8.67 _a_	99.93 ± 0.02^a^	90.95 ± 10.49 ^b^	90.34 ± 6.48 ^bc^	2.45 ± 0.18 ^a^
	*amoA*–AOB	Wheat–Corn	105.0 ± 26.01 ^ab^	99.78 ± 0.08 ^ab^	120.1 ± 22.64 ^ab^	124.0 ± 23.03 ^ab^	3.08 ± 0.37 ^a^
		Soybean–Corn	91.25 ± 24.03 ^ab^	99.81 ± 0.07 ^ab^	105.2 ± 23.30 ^a^	103.9 ± 23.28 ^a^	2.58 ± 0.32 ^a^
		Cotton	89.00 ± 13.60 ^ab^	99.84 ± 0.02 ^b^	104.2 ± 20.83 ^a^	99.9 ± 14.33 ^a^	2.71 ± 0.084 ^a^
		Rice	127.5 ± 30.83 ^b^	99.68 ± 0.09 ^a^	159.7 ± 31.93 ^b^	154.7 ± 30.69 ^b^	2.74 ± 0.33 ^a^
		Fruit	79.25 ± 17.18 ^a^	99.78 ± 0.10 ^ab^	96.9 ± 22.14 ^a^	98.0 ± 28.40 ^a^	2.67 ± 0.68 ^a^
N-fixing bacteria	*nifH*	Wheat–Corn	474.2 ± 182.08 ^a^	99.84 ± 0.04 ^b^	581.3 ± 195.8 ^a^	577.0 ± 164.9 ^a^	4.60 ± 0.38 ^ab^
		Soybean–Corn	411.8 ± 91.25 ^a^	99.84 ± 0.01 ^b^	509.7 ± 80.04 ^a^	489.9 ± 69.41 ^a^	4.37 ± 0.22 ^a^
		Cotton	412.0 ± 27.18 ^a^	99.85 ± 0.03 ^b^	535.1 ± 43.64 ^a^	520.5 ± 62.26 ^a^	4.48 ± 0.40 ^a^
		Rice	998.3 ± 88.85 ^b^	99.46 ± 0.11 ^a^	1187 ± 114.0 ^b^	1163 ± 108.3 ^b^	5.30 ± 0.19 ^b^
		Fruit	450.3 ± 253.0 ^a^	99.84 ± 0.07 ^b^	562.9 ± 240.8 ^a^	570.2 ± 219.1 ^a^	4.33 ± 0.65 ^a^
Denitrifier	*nirK*	Wheat–Corn	434.8 ± 50.92 ^a^	99.90 ± 0.01 ^b^	469.2 ± 51.34 ^a^	455.3 ± 50.21 ^a^	4.53 ± 0.16 ^a^
		Soybean–Corn	391.0 ± 54.54 ^a^	99.89 ± 0.02 ^b^	455.4 ± 35.07 ^a^	423.7 ± 48.51 ^a^	4.40 ± 0.19 ^a^
		Cotton	345.0 ± 48.77 ^a^	99.90 ± 0.00 ^b^	415.8 ± 59.16 ^a^	396.6 ± 54.22 ^a^	4.26 ± 0.42 ^a^
		Rice	649.0 ± 62.52 ^b^	99.70 ± 0.02 ^a^	734.5 ± 82.10 ^b^	722.1 ± 64.21 ^b^	4.43 ± 0.18 ^a^
		Fruit	412.0 ± 114.8 ^a^	99.91 ± 0.02 ^b^	452.0 ± 110.8 ^a^	445.1 ± 105.3 ^a^	4.26 ± 0.51 ^a^
	*nirS*	Wheat–Corn	569.4 ± 107.0 ^b^	99.82 ± 0.05 ^b^	663.7 ± 85.82 ^b^	647.4 ± 92.69 ^b^	4.69 ± 0.09 ^b^
		Soybean–Corn	399.5 ± 20.93 ^a^	99.81 ± 0.04 ^b^	469.4 ± 14.44 ^a^	459.0 ± 20.46 ^a^	3.97 ± 0.25 ^a^
		Cotton	439.0 ± 53.85 ^ab^	99.84 ± 0.04 ^b^	506.7 ± 54.16 ^a^	492.3 ± 53.99 ^a^	4.41 ± 0.29 ^ab^
		Rice	830.0 ± 85.79 ^c^	99.61 ± 0.09 ^a^	951.8 ± 100.7 ^c^	931.0 ± 101.0 ^c^	5.26 ± 0.15 ^c^
		Fruit	425.3 ± 77.04 ^ab^	99.81 ± 0.06 ^b^	515.2 ± 75.98 ^a^	508.1 ± 67.29 ^ab^	4.17 ± 0.56 ^ab^
Bacteria	16S rRNA	Wheat–Corn	2930 ± 85.18 ^a^	98.33 ± 0.28 ^b^	3725 ± 194.1 ^a^	3697 ± 157.9 ^a^	6.49 ± 0.02 ^a^
		Soybean–Corn	3050 ± 323.3 ^a^	98.40 ± 0.13 ^b^	3832 ± 415.1 ^a^	3791 ± 367.1 ^a^	6.50 ± 0.20 ^a^
		Cotton	2929 ± 278.3 ^a^	98.54 ± 0.11 ^b^	3626 ± 314.5 ^a^	3612 ± 314.3 ^a^	6.50 ± 0.16 ^a^
		Rice	3343 ± 279.8 ^a^	97.85 ± 0.15 ^a^	4322 ± 356.0 ^a^	4264 ± 352.4 ^a^	6.71 ± 0.20 ^a^
		Fruit	2957 ± 556.5 ^a^	98.42 ± 0.35 ^b^	3702 ± 648.3 ^a^	3686 ± 676.5 ^a^	6.53 ± 0.25 ^a^

Different letters indicate significant differences based on a Kruskal–Wallis test (*p* < 0.05).

**Table 4 microorganisms-08-00424-t004:** Multi-Response Permutation Procedure (MRPP) results for the effects of agroecosystems on the soil microbial communities: soil bacteria, ammonia oxidizer, denitrifier and N–fixing bacteria.

Microbial Community	Gene Type	*A*	Expected δ	*p*
Ammonia–oxidizers	*amoA*–AOA	0.185	0.426	0.007 **
	*amoA*–AOB	0.018	0.664	0.177
N–fixing bacteria	*nifH*	0.003	0.536	0.433
Denitrifier	*nirK*	0.025	0.525	0.154
	*nirS*	0.063	0.401	0.027 *
Bacteria	16S rRNA	0.532	0.533	0.001 **

The A values are the chance corrected within-group agreements; *p* is the probability of a smaller or equal δ; * significance at the *p* < 0.05 level; ** significance at the *p* < 0.01 level.

**Table 5 microorganisms-08-00424-t005:** The relative abundance of nitrogen functional genera from the five agroecosystems. The genus of AOA–*amoA*, AOB–*amoA*, *nifH*, *nirK*, and *nirS* with relative abundance > 0.01%, > 0.1%, > 0.5%, > 0.5%, > 0.5% at least in one agroecosystem were shown, respectively.

			Corn-Wheat	Cotton	Fruit	Rice	Soybean
Gene	phylum	genus	Relative abundance (%)				
*amoA*–AOA	Thaumarchaeota	*Candidatus_Nitrosocosmicus*	0.05	0.28	0.44	3.41	0.01
	Thaumarchaeota	*Candidatus_Nitrosopelagicus*	0.29	0.03	0.03	0.28	0.04
	Thaumarchaeota	*Candidatus_Nitrosotenuis*	0.14	0.14	0.36	0.33	0.11
	Thaumarchaeota	*Nitrosopumilus*	3.58	4.86	16.05	9.40	3.73
	Thaumarchaeota	*Nitrososphaera*	25.30	27.25	29.86	28.32	29.50
*amoA*–AOB	Basidiomycota	*Coprinopsis*	0.01	0.03	0.02	0.00	0.13
	Proteobacteria	*Anaeromyxobacter*	0.10	0.00	0.00	0.00	0.00
	Proteobacteria	*Roseateles*	0.00	0.00	0.00	1.86	0.00
	Proteobacteria	*Nitrosococcus*	0.02	0.10	0.00	0.01	0.11
	Proteobacteria	*Nitrosomonas*	4.81	11.57	12.16	39.62	1.77
	Proteobacteria	*Nitrospira*	75.65	70.71	69.53	43.19	81.65
	Proteobacteria	*Nitrosovibrio*	7.47	5.62	8.12	2.42	2.14
*nifH*	Proteobacteria	*Burkholderia*	0.77	1.48	0.83	0.08	1.15
	Proteobacteria	*Dechloromonas*	0.56	1.03	1.96	0.84	0.53
	Proteobacteria	*Agrobacterium*	1.47	2.46	6.90	0.08	5.00
	Cyanobacteria	*Anabaena*	0.03	0.59	0.09	0.04	1.40
	Proteobacteria	*Anaeromyxobacter*	0.00	0.00	0.00	0.53	0.01
	Proteobacteria	*Azospira*	0.47	0.75	4.51	0.79	0.21
	Proteobacteria	*Azospirillum*	12.50	14.25	8.93	1.72	20.11
	Proteobacteria	*Azotobacter*	0.52	1.26	5.44	1.29	1.12
	Proteobacteria	*Bradyrhizobium*	3.43	2.30	2.22	6.21	2.93
	Proteobacteria	*Derxia*	0.00	0.00	0.00	1.05	0.00
	Proteobacteria	*Desulfobulbus*	0.19	0.41	1.14	4.73	0.00
	Proteobacteria	*Desulfovibrio*	1.13	1.67	1.59	8.98	0.60
	Proteobacteria	*Desulfuromonas*	3.17	5.67	1.99	7.68	1.61
	Proteobacteria	*Geobacter*	4.31	6.92	3.26	4.33	5.40
	Proteobacteria	*Paraburkholderia*	9.06	4.25	9.18	1.19	5.25
	Proteobacteria	*Pseudacidovorax*	4.16	0.44	0.43	2.44	0.46
	Proteobacteria	*Pseudomonas*	3.55	1.01	1.93	4.17	1.39
	Proteobacteria	*Rhizobium*	0.46	0.44	1.08	0.01	0.79
	Proteobacteria	*Rubrivivax*	0.49	0.58	0.89	0.44	0.82
	Proteobacteria	*Sideroxydans*	0.94	0.55	0.05	1.41	0.04
	Proteobacteria	*Stenotrophomonas*	0.01	0.80	0.04	0.06	0.26
	Proteobacteria	*Skermanella*	4.93	5.53	4.37	0.20	7.08
	Cyanobacteria	*Leptolyngbya*	0.00	0.00	0.00	1.50	0.00
	Cyanobacteria	*Nostoc*	0.63	0.00	0.00	0.00	0.00
	Firmicutes	*Pelosinus*	0.00	0.00	0.00	1.00	0.00
*nirK*	Proteobacteria	*Alcaligenes*	4.16	0.66	0.25	3.64	0.49
	Proteobacteria	*Bosea*	2.85	1.73	0.14	0.75	0.61
	Proteobacteria	*Bradyrhizobium*	15.48	13.10	10.66	11.04	15.91
	Proteobacteria	*Chelativorans*	4.56	7.00	0.90	1.11	5.66
	Proteobacteria	*Chelatococcus*	1.08	3.66	2.87	0.77	1.97
	Proteobacteria	*Enterobacter*	0.51	0.18	0.50	1.62	0.26
	Proteobacteria	*Mesorhizobium*	13.97	20.40	19.78	19.34	12.51
	Proteobacteria	*Nitrosomonas*	0.29	0.00	0.00	4.59	0.11
	Proteobacteria	*Ochrobactrum*	2.71	0.00	5.90	0.95	0.05
	Proteobacteria	*Rhizobium*	11.56	6.47	8.76	29.13	16.84
	Proteobacteria	*Rhodobacter*	0.03	0.00	0.02	1.91	0.00
	Proteobacteria	*Rhodopseudomonas*	13.91	6.65	11.77	2.99	14.90
	Proteobacteria	*Sinorhizobium*	9.64	21.05	22.90	6.16	13.41
*nirS*	Proteobacteria	*Alicycliphilus*	0.00	0.00	0.00	0.53	0.00
	Proteobacteria	*Aromatoleum*	0.03	0.00	0.00	0.89	0.00
	Actinobacteria	*Arthrobacter*	0.00	0.00	0.50	0.00	0.00
	Proteobacteria	*Azoarcus*	6.07	3.80	9.31	13.05	3.32
	Proteobacteria	*Azospira*	0.53	0.01	0.54	0.11	0.08
	Proteobacteria	*Azospirillum*	3.81	4.31	4.74	0.87	7.00
	Firmicutes	*Bacillus*	0.97	0.22	0.72	1.58	0.04
	Proteobacteria	*Cupriavidus*	6.89	6.20	3.65	2.25	5.05
	Proteobacteria	*Halomonas*	0.08	0.58	0.89	0.03	0.04
	Proteobacteria	*Magnetospirillum*	2.41	1.83	3.81	0.42	5.63
	Proteobacteria	*Paracoccus*	0.60	1.55	0.01	0.25	0.01
	Proteobacteria	*Polymorphum*	0.85	0.32	1.04	0.50	0.61
	Proteobacteria	*Pseudogulbenkiania*	0.05	0.00	0.00	1.95	0.00
	Proteobacteria	*Pseudomonas*	3.88	2.81	3.44	9.72	0.77
	Proteobacteria	*Rhodanobacter*	5.01	3.26	4.98	3.99	16.94
	Proteobacteria	*Rubrivivax*	5.61	4.12	4.03	1.57	5.49
	Proteobacteria	*Sulfuritalea*	6.82	3.40	2.28	5.31	1.89
	Proteobacteria	*Thauera*	0.05	0.00	0.00	0.79	0.00
	Proteobacteria	*Thiobacillus*	0.75	0.02	0.00	0.66	0.00

**Table 6 microorganisms-08-00424-t006:** Results for RDA testing effects of soil properties on the community structures of ammonia–oxidizers (*amoA*), denitrifying bacteria (*nirS* and *nirK*), N–fixing bacteria (*nifH*), and total bacteria (16S rRNA).

Soil Properties	Degrees of Explanation (%)			
	AOA–*amoA* and AOB–*amoA*	*nirS* and *nirK*	*nifH*	16S rRNA
SWC	45.4 **	21.6 **	54.2 **	38.5 **
EC	15.3 **	13.7 **	NS	8.9 **
TN	9.7 *	NS	NS	NS
AP	NS	9.1 *	NS	21.5 **
CEC	NS	NS	5.8 *	NS
NO_3_^−^–N	NS	NS	NS	NS

SWC, AP, CEC, EC, TN, and NO_3_^−^–N represent the water content, available phosphorous, effective cation exchange capacity, electrical conductivity, total nitrogen, and nitrate contents, respectively. * significant at the *p* < 0.05 level; ** significant at the *p* < 0.01 level; NS represents no significant.

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
