# Peer review of "The Structure and Diversity of Nitrogen Functional Groups from Different Cropping Systems in Yellow River Delta"

_microorganisms, 2020, doi:10.3390/microorganisms8030424_

Round 1

Reviewer 1 Report

The submitted manuscript entitled "The structure and diversity of nitrogen functional groups from different cropping systems in Yellow River Delta" seems to be an adequate for Microorganisms Journal.

There are some minor comments which have to be taken into account by Authors and corrected.

Abstract

L.14-19. the first sentence in Abstract is too long and complicated. Please rewrite it.
L.14-15. «To study some steps of N cycle process» - That’s not true. In your study, you didn’t consider steps of N cycle process, but determined abundance and diversity of N-cycle microorganisms. Please, correct.

L.26-28. your final conclusion concerning «the local farmers should think highly of optimizing management of soil water and nitrogen fertilizer in agroecosystem of YRD» is overconfident and not supported enough by the data and therefore should be fully rewritten. The number of functional genes is very often not related to the activity of microbial processes. For this kind of conclusions, you had to measure not only functional genes abundances, but also nitrogen fixation and denitrification activity and many-many other soil properties. Thus, your conclusion should be about the effect of environmental factors on N-cycles microorganisms, but not about the farmers.

Highlights

«Rice had higher diversity of bacterial and targeted N-cycling microbial community» - Soil under Rice but, not Rice. Please, correct.

Introduction
L.51. What about N2?

L.55-56. I am not sure that the use of diazotrophic bacteria is efficient strategy to an efficient strategy for improving the N use efficiency since in agroecosystems when added N fertilizers the activity of nifH genes are fully suppressed. Moreover, the addition of microbes into soil showed its inefficiency to improve soil characteristics.
L. 56-58.  It’s the first results based on scientific study, so please avoid mentioning farmers in this context.

L.67-68. «some steps of soil N-cycling microbes» - please rewrite
L. 80-82. «This study could help local famers choose right cropping strategy to improve the N fertilizer use efficiency and reduce N loss.» - Please remove this sentence. Simple measuring of abundance or diversity of soil microorganisms cannot help to choose right cropping strategy. It can only help us to have an insight to some soil ecology patterns related to microbial communities in soil.

Discussion
L. 374. High-throughput sequencing

L.373-376. in these sentences there are a lot of repetitions of text mentioned in introduction and M&M. Please rewrite it.

Author Response

Response to Reviewer 2 Comments

The submitted manuscript entitled "The structure and diversity of nitrogen functional groups from different cropping systems in Yellow River Delta" seems to be an adequate for Microorganisms Journal.

There are some minor comments which have to be taken into account by Authors and corrected.

Abstract

Point 1: L.14-19. The first sentence in Abstract is too long and complicated. Please rewrite it.

Response 1: Thank you for the suggestion. We have rewritten the sentence, as following: “The Yellow River Delta (YRD) region is an important production base in Shandong Province. It encompasses an array of diversified crop systems, including corn-wheat rotation system (Wheat-Corn), soybean-corn rotation system (Soybean-Corn), fruits or vegetables system (Fruit), Cotton system (Cotton) and Rice system (Rice). In this study, the communities of ammonia oxidizer, denitrifier and nitrogen (N)-fixing bacteria in those cropping systems were investigated by Illumina Miseq sequencing.

Point 2: L.14-15. «To study some steps of N cycle process» - That’s not true. In your study, you didn’t consider steps of N cycle process, but determined abundance and diversity of N-cycle microorganisms. Please, correct.

Response 2: Thank you for the suggestion. We have corrected it accordingly as following: “In this study, the communities of ammonia oxidizer, denitrifier and nitrogen (N)-fixing bacteria in those cropping systems were investigated by Illumina Miseq sequencing.”

Point 3: L.26-28. your final conclusion concerning «the local farmers should think highly of optimizing management of soil water and nitrogen fertilizer in agroecosystem of YRD» is overconfident and not supported enough by the data and therefore should be fully rewritten. The number of functional genes is very often not related to the activity of microbial processes. For this kind of conclusions, you had to measure not only functional genes abundances, but also nitrogen fixation and denitrification activity and many-many other soil properties. Thus, your conclusion should be about the effect of environmental factors on N-cycles microorganisms, but not about the farmers.

Response 3: Thank you for the suggestion. We totally agree with reviewer. We have removed the conclusion that our results provide farmers instructions on cropping strategy.

Highlights

Point 4: «Rice had higher diversity of bacterial and targeted N-cycling microbial community» - Soil under Rice but, not Rice. Please, correct.

Response 4: Thank you for reviewer’s suggestion. We have corrected it accordingly as following: Rice soil had higher diversity of studied N-cycling microbial guilds.

Introduction
Point 5: L.51. What about N2?

Response 5: Thank you for reviewer’s question. Here, we have carefully checked the reference, and rewritten this sentence as following: “Ammonia-oxidizer and nitrifier oxidize ammonia into nitrite and then nitrate, whilst the denitrifier reduces nitrate into nitrite, nitrous oxide, nitric oxide, and finally nitrogen gas”.

Point 6:L.55-56. I am not sure that the use of diazotrophic bacteria is efficient strategy to an efficient strategy for improving the N use efficiency since in agroecosystems when added N fertilizers the activity of nifH genes are fully suppressed. Moreover, the addition of microbes into soil showed its inefficiency to improve soil characteristics.

Response 6: Thank you for the suggestion. We agree with that whether addition of diazotrophic bacteria into soil would improve N use efficiency is unclear. We have changed the sentence into “In intercropping systems, the nitrogen activeness of soybean nodule bacterium can reduce N fertilizer input and consequently increase N use efficiency”.

Point 7:L. 56-58.  It’s the first results based on scientific study, so please avoid mentioning farmers in this context.

Response 7: Thank you for reviewer’s suggestion. We totally agree with reviewer. And we have rewritten this sentence, as following: Thus, suppressing ammonia-oxidizer growth was shown to increase N use efficiency of potato tubers [14].

Point 8: L.67-68. «some steps of soil N-cycling microbes» - please rewrite

Response 8: Thank you for reviewer’s suggestion. And we have rewritten this sentence, as following: “In this study, we aimed to investigate how different cropping systems affect the communities of ammonia oxidizer, denitrifier and nitrogen (N)-fixing bacteria”.

Point 9:L. 80-82. «This study could help local famers choose right cropping strategy to improve the N fertilizer use efficiency and reduce N loss.» - Please remove this sentence. Simple measuring of abundance or diversity of soil microorganisms cannot help to choose right cropping strategy. It can only help us to have an insight to some soil ecology patterns related to microbial communities in soil.

Response 9: Thank you for reviewer’s suggestion. We have removed this sentence.

Discussion
Point 10:L. 374. High-throughput sequencing

Response 10: Thank you for reviewer’s suggestion. We have corrected it accordingly.

Point 11:L.373-376. In these sentences there are a lot of repetitions of text mentioned in introduction and M&M. Please rewrite it.

Response 11: Thank you for reviewer’s suggestion. We have rewritten these sentences, as following, “In this study, we investigated the communities of soil bacteria, ammonia oxidizer, denitrifier and nitrogen-fixing bacteria by high throughput sequencing”.

Reviewer 2 Report

The manuscript from He et al. considers some steps in the nitrogen cycle in five agricultural systems. In detail, genes from nifH nitrogenase, amoA nitrification and nirK and nirS denitrification were studied. The aim was to detecte differences among agricultural systems and their relation with important soil facts that might affect the microbial community in the soil.
The manuscript is very interesting, the data are well presented and discussed. I think that figures 7 & 8 are not easy to understand for those who are not accostumed to the specific statistical analysis. Thus, I would like to ask the Authors for a bit of extra explanantion in the text about this analysis. I also suggest that the Authors have their manuscript read by a native English speaker, as the text is not always fluid.

In conclusion, I think the manuscript can be accepted after minor revisions.

Author Response

Response to Reviewer 1 Comments

Comments and Suggestions for Authors

The manuscript from He et al. considers some steps in the nitrogen cycle in five agricultural systems. In detail, genes from nifH nitrogenase, amoA nitrification and nirK and nirS denitrification were studied. The aim was to detecte differences among agricultural systems and their relation with important soil facts that might affect the microbial community in the soil.

The manuscript is very interesting; the data are well presented and discussed. In conclusion, I think the manuscript can be accepted after minor revisions.

Point 1: I think that figures 7 & 8 are not easy to understand for those who are not accustomed to the specific statistical analysis. Thus, I would like to ask the Authors for a bit of extra explanation in the text about this analysis.

Response 1: Thank you for the suggestion. We have explained how to make networks, and added the information in revised manuscript (Materials and methods section, 2.7 High-throughput sequencing and Data analysis), as following:

“Firstly, we defined the “dominant” OTUs with relative abundances > 0.5% within at least one replicate in each cropping system. Then, the matrix was established based on the Pearson correlation coefficient using the relative abundance of each “dominant” OTU. The network was conducted through employing a threshold (Pearson correlation coefficient |r| >0.85, p <0.05). The network structure was visualized with Cytoscape (version 3.6.0). In the network structure, node values represent the relative abundances of OTUs, and the edge sizes indicate the Pearson correlation coefficients between each two OTUs.”

Point 2: I also suggest that the Authors have their manuscript read by a native English speaker, as the text is not always fluid.

Response 2: Thank you for reviewer’s suggestion. The manuscript has been edited by a native English speaker. This is the certificate.

(the certificate is in the word version)

This manuscript is a resubmission of an earlier submission. The following is a list of the peer review reports and author responses from that submission.

Round 1

Reviewer 1 Report

It was my sincerely pleasure to review the manuscript entitled: “The structure and diversity of nitrogen functional prokaryotic groups from different cropping systems”.

The work is very well written.

Authors used sequencing technique to decipher the bacterial composition in five different cropping systems. Additionally, the qPCR technique was utilized to quantify five different genes involved in N-turnaround. The same genes were also sequenced using high-throughput sequencing.

All data were analyzed using innovative, up-to-date and sophisticated bioinformatics and statistical methods.

My only objection is that the research and the manuscript are of local interest, as the results are not scalable or comparable to other ecosystems or cropping systems in different locations.

I have no other major comments, only some minor suggestions, listed below:

ln:177: It is highly recommended to cluster OTUs with at least 99% similarity cutoff

ln 195: „samples”, should be „sample”

Figure 2: Please use the same colors in legend as in figure

Table 6: Should be placed in supplementary material, since it doesn’t bring any necessary information

Figure 9: This figure is low quality, especially labels are too small and fuzzy

ln 389: Your sequencing data is presented as taxonomy down to the genus level, so you cannot say “dominant species” etc.

Author Response

Response to Reviewer 1Comments

Point 1: It was my sincerely pleasure to review the manuscript entitled: “The structure and diversity of nitrogen functional prokaryotic groups from different cropping systems”.

The work is very well written.

Authors used a sequencing technique to decipher the bacterial composition in five different cropping systems. Additionally, the qPCR technique was utilized to quantify five different genes involved in N-turnaround. The same genes were also sequenced using high-throughput sequencing. All data were analyzed using innovative, up-to-date and sophisticated bioinformatics and statistical methods.

Response 1:  We are fully grateful for this comment. It is worthy recognition of our research and encouragement.

Point 2: My only objection is that the research and the manuscript are of local interest, as the results are not scalable or comparable to other ecosystems or cropping systems in different locations.

Response 2: We agree with the comment that our manuscript is of local interest. Therefore, we changed the title as “The structure and diversity of nitrogen functional groups from different cropping systems in Yellow River Delta”.

The main purpose of this manuscript is to reveal how different cropping systems affect soil N-cycling microbes in Yellow River Delta, which is one of the fastest-growing deltas in the world and being developed into an important agricultural production base in China. The environmental background and crop systems could be unique in the Yellow River Delta. Nevertheless, analysis of N-cycling microbes is still of special importance to improve N fertilizer use efficiency and reduce N loss for agricultural sustainable development in YRD.

At the same time, we have modified the last paragraph of the introduction part, in order to clarify our research purpose, as follows: In this study, we aimed to investigate how different cropping systems affect some steps of soil N-cycling microbes in YRD. Crop systems of YRD have been progressively converted from continuous cotton to soybean system and then to wheat-corn rotation in the past two decades, and now, the cropping systems in YRD are diversified, including winter wheat-summer maize rotation system (Wheat-Corn), continuous cotton system (Cotton), and continuous rice (Rice), soybean-maize rotation system (Soybean-Corn), fruits and vegetable system (Fruit). We collected soil samples from the five cropping systems, Wheat-Corn, Soybean-Corn, Cotton, Rice, Fruit-Vegetable, to study the communities of total bacteria and N-cycling microbes based on the Illumina MiSeq sequencing of functional genes. We used representative primer sets to amplify 16S rDNA for soil total bacteria, nifH for targeting biological N-fixation bacteria, nirS and nirK for denitrifying bacteria, and amoA for ammonia-oxidization archaea (AOA) and ammonia-oxidization bacteria (AOB), respectively. Two objectives in our study were pursued: (1) the effect of different cropping systems in YRD on soil bacterial and targeted N-cycling functional groups; (2) the contribution of soil properties to the differences in targeted N-cycling functional groups in the five cropping systems. This study could help local farmers choose the right cropping strategy to improve the N fertilizer use efficiency and reduce N loss.(In INTRODUCTION part, Lines 68-83)

I have no other major comments, only some minor suggestions, listed below:

Point 3: ln:177: It is highly recommended to cluster OTUs with at least 99% similarity cutoff

Response 3: We totally agree with the reviewer that the UPARSE pipeline with 99% similarity would lead to OTUs being closer to true biological species. In our study, we used 97% similarity to pick OTUs and the similarity cutoff seemed to be quite acceptable. First, the microbial composition between groups was observed to be significantly different between groups based on our cutoff. It indicates that our similarity cutoff successfully detected the microbial divergence between groups.  Second, our results were shown at either phylum or genus level, therefore they did not demand a high resolution at OTU picking. Third, we subjectively suspect that a higher similarity cutoff would not lead to a different conclusion in this study since the divergence between groups was large. Last, we appreciate the reviewer's recommendation and we will use 99% and even 100% similarity cutoff in our future study.

Point 4: ln 195: „samples”, should be „sample”

Response 4: We have corrected accordingly as the reviewer suggested.

Point 5: Figure 2: Please use the same colours in legend as in figure

Response 5: Thanks for the suggestion. We have used the same colours in legend as in figure 2.

Point 6: Table 6: Should be placed in supplementary material, since it doesn’t bring any necessary information

Response 6: Thanks for the suggestion. We have removed Table 6 to the supplementary material part.

Point 7: Figure 9: This figure is low quality; especially labels are too small and fuzzy

Response 7: We have modified Figure 9 and the new figure is more readable.

Point 8: ln 389: Your sequencing data is presented as taxonomy down to the genus level, so you cannot say “dominant species” etc.

Response 8: We totally agree with the reviewer’s suggestion. We think “organisms” is more appropriate here. So, we changed the “dominant species” as “dominant organisms”.

Reviewer 2 Report

The paper entitled “The structure and diversity of nitrogen functional 2 prokaryotic groups from different cropping systems” by He et al. study some steps of the nitrogen cycle through estimate the abundances of 16S rRNA, nifH, amoA,nirS andnirK genesassociated to soils of five cropping systems in Yellow River Delta. The sampling was well designed, the experimental part and analysis of the data are well chosen, and the figures, tables and supplementary file are relevant. Nevertheless, the work should be reviewed very carefully as it is full of types, some examples of which appear in Minor details. The conclusions seem to be a continuation of the discussion and should be rewritten succinctly and briefly. Also the discussion section should be significantly improved.

Suggested elements should be taken for the discussion of the work

The authors must recognize that only genetic markers of some steps of the nitrogen cycle were included. For example, the ANAMOX step was not explored. Nor should it be mentioned in any section that the diversity of prokaryotes was explored, because the molecular targets chosen are strictly bacterial and no archaea was included. The authors pick up oligonucleotide designs to gene amplifications reported in the literature dating back many years (1997, 1999, 2000, 2005, 2009), when the databases were much more limited and the known bacterial diversity was very limited. In my opinion, the authors had to make new designs of "universal" oligonucleotides with the information available at this time. The authors have to study what taxa these oligonucleotides are expected to recognize and which taxonomic groups are theoretically left out. In all sections it should be noted which taxonomic groups can theoretically be amplified, but also warn of taxa that do not fall within this experimental design. Without this study and warnings, I don't see how to accept the paper, because this limitation will lead to new mistakes in other future papers. At least in the case of the NifH genes, I am sure that the oligos used only amplify a very small fraction of this very diverse superfamily of genes in the broad phylogenetic tree of bacteria. There is no universal gene for NifK genes, due to the enormous diversity currently recognized. Some conserved regions could allow the design of oligonucleotides, but they would have so many "degenerations" that they become useless for mass sequencing methods. The authors must recognize the limitations of this part of their experimental design, warn of the limitations of their oligonucleotides and discuss the results based on the taxa found, recognizing that many other taxa could not be detected given the experimental design chosen and the immense diversity of the genes that were chosen as target. The diversity of AOA and denitrifiers is also underestimated and this should be recognized and warned to readers. I am clear that the abundance or diversity of AOA genes was different in the different soils and that too obvious differences were not observed with the other functional markers, however those guilds (nitrogen fixers and denitrifiers) or total bacteria should also be discussed. No one doubts the importance of the information produced by the methods of mass sequencing, but the work would have had a greater contribution if phenotypic methods or tests (reduction of acetylene, ammonium oxidation, denitrification) had been used to relate the nitrogen cycle with the abundance of organisms that participate in the cycle in each of the soils.Finally, the metagenomic DNA is a "photograph" of the organisms present in a sample, but it does not allow recognizing those that are viable of those that are not; it does not allow to recognize the metabolically active groups of those in a dormant state. I think this also has to be recognized.

Minor details (non-extensive registration)

Page 1, line 18 change “prokaryotic” by “bacteria”

Page 1, lines 14-15, change “To study the whole 14N cycle process” by “To study some steps of N cycle process”
In all the work separate the magnitude of the units. For example:

Page 4, line 133 Change “20ml” by “20 mL”

Page 4, line 134 Change “0.2g/L” by “0.2 g/L”

Page 5, Line 165 Change “5xBuffer” by “5x Buffer”

Table 2. Change “SWC, AP, AK, pH, SOM, TN, NH4+-N, NO3--N, CEC, and EC represent the water content, available phosphorous, available potassium, pH, organic matter, total 210 nitrogen, ammonium, nitrate, effective cation exchange capacity, and electrical conductivity of soil samples, respectively” by “SWC, water content; AP, available phosphorous; AK, available potassium; SOM, organic matter; TN, total nitrogen; NH4+-N, ammonium; NO3--N, nitrate; CEC, effective cation exchange capacity; and EC, electrical conductivity.

Figure 2. Change “ammoniz –oxidizers” by “ammonia –oxidizers”

Table 7. Change “16s rRNA” by “16S rRNA”

Author Response

Response to Reviewer 2 Comments

Point 1: Nevertheless, the work should be reviewed very carefully as it is full of types, some examples of which appear in Minor details. The conclusions seem to be a continuation of the discussion and should be rewritten succinctly and briefly.

Response 1: Thank you for reviewer’s comment. We have re-written succinctly and briefly the conclusion part and made the conclusion part more readable,

as following: Our research clarified the effect of different cropping systems in the YRD on the abundance and diversity of selected N functional genes, including nitrification gene amoA, denitrification genes nirK and nirS, and nitrogenase gene nifH, and explored the contribution of soil properties to the abundance and diversity of these genes. We found that (1) Rice soil had significantly higher diversity indices of soil bacterial and targeted N-cycling functional communities than other crop soils; (2) Rice soils with higher SWC had lower amoA abundance and soil PNR, indicating Rice soils had lower abundance and activity of ammonia oxidizer; (3) Wheat-Corn soil with higher level of fertilization had higher abundance of nitrification gene amoA and denitrification genes nirK and nirS, compared with Soybean-Corn, Cotton and Fruit soils; (4) SWC, EC, and TN were the most important influencing factors of soil bacterial and targeted N-cycling microbial diversity and structure. Additionally, considering that compound fertilizer 600 kg h-1year-1 and diammonium hydrogen phosphate 550 kg h-1year-1 were used in Wheat-Corn soils, it is urgent to take actions to improve the N fertilizer use efficiency and reduce N loss, for example, optimizing soil water management and nitrogen fertilizer.”

Point 2:Also the discussion section should be significantly improved.Suggested elements should be taken for the discussion of the work.

The authors must recognize that only genetic markers of some steps of the nitrogen cycle were included. For example, the ANAMOX step was not explored. Nor should it be mentioned in any section that the diversity of prokaryotes was explored, because the molecular targets chosen are strictly bacterial and no archaea was included.

Response 2: We totally agree with reviewer. Indeed, we only focused on the abundance and diversity of selected N functional genes, including nitrification gene amoA, denitrification genes nirK and nirS, and nitrogenase gene nifH, which are involved in ammonia oxidation, denitrification and nitrogen fixation, respectively. We have also modified our manuscript to make that point clearly, as following:

To study some steps of N cycle process in five cropping systems in Yellow River Delta (YRD), including corn-wheat rotation system (Wheat-Corn), soybean-corn rotation system (Soybean-Corn), fruits or vegetables system (Fruit), Cotton system (Cotton) and Rice system (Rice), nitrification gene amoA, denitrification genes nirK and nirS, and nitrogenase gene nifH were investigated by the Illumina MiSeq sequencing (in ABSTRACT part, Lines 15-20). In this study, we aimed to investigate how different cropping systems affect some steps of soil N-cycling microbes in YRD (in INTRODUCTION part, Lines 68-69). Our research clarified the effect of different cropping systems in the YRD on the abundance and diversity of selected N functional genes, including nitrification gene amoA, denitrification genes nirK and nirS, and nitrogenase gene nifH, and explored the contribution of soil properties to the abundance and diversity of these genes (in CONCLUSIONS part, Lines 464-466).

Point 3: The authors pick up oligonucleotide designs to gene amplifications reported in the literature dating back many years (1997, 1999, 2000, 2005, 2009), when the databases were much more limited and the known bacterial diversity was very limited. In my opinion, the authors had to make new designs of "universal" oligonucleotides with the information available at this time. The authors have to study what taxa these oligonucleotides are expected to recognize and which taxonomic groups are theoretically left out. In all sections it should be noted which taxonomic groups can theoretically be amplified, but also warn of taxa that do not fall within this experimental design. Without this study and warnings, I don't see how to accept the paper, because this limitation will lead to new mistakes in other future papers. At least in the case of the NifH genes, I am sure that the oligos used only amplify a very small fraction of this very diverse superfamily of genes in the broad phylogenetic tree of bacteria. There is no universal gene for NirK genes, due to the enormous diversity currently recognized. Some conserved regions could allow the design of oligonucleotides, but they would have so many "degenerations" that they become useless for mass sequencing methods. The authors must recognize the limitations of this part of their experimental design, warn of the limitations of their oligonucleotides and discuss the results based on the taxa found, recognizing that many other taxa could not be detected given the experimental design chosen and the immense diversity of the genes that were chosen as target. The diversity of AOA and denitrifiers is also underestimated and this should be recognized and warned to readers.

Response 3: Thanks a lot for the comment of reviewer. In the original manuscript, we did ignore the limitation of primer sets. And, we have not warned which taxa would not fall within our experimental design. In our revised manuscript, we recognized that the sequencing method only reveals the presence of organisms containing these genes but cannot reveal whether they are viable or metabolically active. In addition, some organisms could have been ignored due to the limitation of primer sets used in the study. Therefore, we have re-written the discussion part, adding the limitation of experimental design and warning which taxa would not be detected under the used primer sets, as following (in DISCUSSION part, Lines 375-394):

“In this study, we investigate the diversity and abundance of four N-cycling functional genes using PCR with the listed primers (Table 1) and high throughout sequencing. The chosen genes include nitrification gene amoA, denitrification genes nirK and nirS, and nitrogenase gene nifH, which are involved in ammonia oxidation, denitrification and nitrogen fixation, respectively [15, 29]. This method only reveals the presence of organisms containing these genes but cannot reveal whether they are viable or metabolically active. In addition, some organisms could have been ignored due to the limitation of primer sets used in the study.

We chose the nifH gene because it contains more conserved regions than other nitrogenase genes nifD and nifK and has been widely used to study diazotroph groups [30]. Nevertheless, the Poly primer set used in our study cannot identify the cluster IV/V of nifH as revealed by a latest study [31]. Therefore, the diversity of diazotrophs should have been underestimated in our study. The nirK and nirS genes were analyzed by the conventional primers F1aCu/R3Cu and cd3aF/R3cd, respectively. The primers amplify the nir genes in Cluster I, including nirK from the class alpha-, beta- and gamma-proteobacteria, and nirS from the class alpha-, beta-, gamma- and epsilon-proteobacteria, and the phyla Bacteroidetes, Planctomycetes and NC10 [27]. However, the primer sets fail to amplify the Clusters II–IV of nir genes [27]. Primers used to investigate bacterial and archaeal amoA cannot cover all AOA and AOB, either. For example, the primer set Arch amoA-F/Arch amoA-R only covers 37% Nitrosotalea subclade 1.1 members but not Nitrosotalea subclade 2 in agricultural soils [32]. The primer amoA-1F/ amoA-2R is limited for revealing the Nitrosospira 8A clade [32].”

Point 4: I am clear that the abundance or diversity of AOA genes was different in the different soils and that too obvious differences were not observed with the other functional markers, however those guilds (nitrogen fixers and denitrifiers) or total bacteria should also be discussed.

Response 4: Thanks a lot for the comment of reviewer. In the revised manuscript, we have discussed the difference of nitrogen fixer and denitrifier in different cropping systems, as following:

Moreover, the Wheat-Corn rotation soil is also featured with high abundance of denitrification genes (nirK and nirS) and network density of nirS, suggesting that it contained more denitrification microbes.” (In DISCUSSION part, Lines 403-405).

 “The Rice soil contained the most abundant nitrogenase gene nifH and denitrification genes (nirK and nirS), which could be due to high water content (more discussion in section 4.2).” (In DISCUSSION part, Lines 418-419)

 “For diazotrophs, nifH abundance was higher in Rice soils than that in other cropping soils in our study (p < 0.05) (Fig. 5B), and the SWC was positively related to N fixing bacterial community. A reason for the phenomena could be that higher SWC can enhance the diffusive transport of organic substrates, which might indirectly increase the access of organic C substrate to N fixing bacteria.” (In DISCUSSION part, Lines 440-444)

Point 5: No one doubts the importance of the information produced by the methods of mass sequencing, but the work would have had a greater contribution if phenotypic methods or tests (reduction of acetylene, ammonium oxidation, denitrification) had been used to relate the nitrogen cycle with the abundance of organisms that participate in the cycle in each of the soils.

Response 5: Thanks a lot for the comment of reviewer. We have added that information in revised manuscript, as following: “Moreover, next study should be focused on the relationship between the diversity and abundance of N-cycling microbial community and N cycle activities, such as ammonium oxidation, denitrification and N fixation in the crop systems to reveal the contribution of N-cycling microbes to N cycle.” (In DISCUSSION part, Lines 459-462)

Point 6: Finally, the metagenomic DNA is a "photograph" of the organisms present in a sample, but it does not allow recognizing those that are viable of those that are not; it does not allow recognizing the metabolically active groups of those in a dormant state. I think this also has to be recognized.

Response 6: Thanks a lot for the comment of reviewer. We totally agree with reviewer. And, we have added this information in revised manuscript, as following (In DISCUSSION part, Lines 379-380):

“This method only reveals the presence of organisms containing these genes but cannot reveal whether they are viable or metabolically active.”

Minor details (non-extensive registration)

Point 7: Page 1, line 18 change “prokaryotic” by “bacteria”

Response7: Thank you for reviewer’s suggestion. We have rewritten the sentence, as following:

“To study some steps of N cycle process in five cropping systems in Yellow River Delta (YRD), including corn-wheat rotation system (Wheat-Corn), soybean-corn rotation system (Soybean-Corn), fruits or vegetables system (Fruit), Cotton system (Cotton) and Rice system (Rice), nitrification gene amoA, denitrification genes nirK and nirS, and nitrogenase gene nifH were investigated by the Illumina MiSeq sequencing.”

Point 8:Page 1, lines 14-15, change “To study the whole 14N cycle process” by “To study some steps of N cycle process”

Response 8: Thank you for reviewer’s suggestion. We have changed it accordingly.

Point 9: In all the work separate the magnitude of the units. For example:Page 4, line 133 Change “20ml” by “20 mL”ï¼›Page 4, line 134 Change “0.2g/L” by “0.2 g/L”ï¼›Page 5, Line 165 Change “5xBuffer” by “5x Buffer”

Response 9: Thank you for reviewer’s suggestion. We have changed it accordingly.

Point 10:Table 2. Change “SWC, AP, AK, pH, SOM, TN, NH4+-N, NO3--N, CEC, and EC represent the water content, available phosphorous, available potassium, pH, organic matter, total nitrogen, ammonium, nitrate, effective cation exchange capacity, and electrical conductivity of soil samples, respectively” by “SWC, water content; AP, available phosphorous; AK, available potassium; SOM, organic matter; TN, total nitrogen; NH4+-N, ammonium; NO3--N, nitrate; CEC, effective cation exchange capacity; and EC, electrical conductivity.

Response 10: Thank you for reviewer’s suggestion. We have changed it accordingly.

Point 11:Figure 2. Change “ammoniz –oxidizers” by “ammonia –oxidizers”

Response 11: Thank you for reviewer’s suggestion. We have changed it accordingly.

Point 12:Table 7. Change “16s rRNA” by “16S rRNA”

Response 12: Thank you for reviewer’s suggestion. We have changed it accordingly.

Round 2

Reviewer 2 Report

The authors made an effort to answer all the suggestions and criticisms of the work point by point, and also recognized the limitations of the work derived from the methods used and the complexity of the bacterial guilds studied. In my opinion, work can be accepted.